# Association Between Body Image and Quality of Life of Women Who Underwent Breast Cancer Surgery

**DOI:** 10.3390/ijerph22071114

**Published:** 2025-07-15

**Authors:** Camila Zanella Battistello, Eduardo Remor, Ícaro Moreira Costa, Mônica Echeverria de Oliveira, Andréa Pires Souto Damin

**Affiliations:** 1Postgraduate Program in Psychology, Universidade Federal do Rio Grande do Sul, Porto Alegre 90035-003, Brazil; czbattistello@gmail.com; 2Health Sciences Center, Universidade de Fortaleza, Fortaleza 60811-905, Brazil; 3Psychology Service, Hospital de Clinicas de Porto Alegre, Porto Alegre 90035-903, Brazil; 4Mastology Service, Hospital de Clinicas de Porto Alegre, Porto Alegre 90035-903, Brazil; 5Postgraduate Program in Health Sciences: Gynecology and Obstetrics, Universidade Federal do Rio Grande do Sul, Porto Alegre 90035-003, Brazil

**Keywords:** body image, breast cancer, oncology, quality of life, surgical treatment

## Abstract

Breast cancer is a condition characterized by the uncontrolled growth of breast cancer cells. The treatment for the disease, such as surgery, chemotherapy, radiotherapy, and systemic therapy, can significantly impact patients’ body image and overall quality of life. This study aimed to evaluate body image perceptions and cancer-related quality of life in women who underwent surgical treatment for breast cancer at a reference hospital in southern Brazil. One hundred six women with breast cancer, aged 21 to 93 years (*M* = 55.3; *SD* = 12.9), participated in this cross-sectional study. They responded to the Body Image and Relationships Scale (BIRS), Functional Assessment of Cancer Therapy for Breast Cancer scale (FACT-B), and a questionnaire on clinical and sociodemographic variables. Multiple linear regression analyses revealed that general perceived body image, as measured by BIRS, was significantly predicted by younger age and chemotherapy (*F*(2, 99) = 7.376, *p* = 0.003). These predictors accounted for 11.2% of the variance in BIRS (adjusted *R*^2^ = 0.112). Hierarchical multiple regression analysis indicated that cancer-related quality of life was significantly predicted by younger age, use of psychiatric medication, and body image domains, including strength and health, social barriers, and appearance and sexuality. The complete model, encompassing all predictors, was significant (*F*(5, 96) = 15.970, *p* < 0.001) and explained 42.6% of the variance in FACT-B (adjusted *R*^2^ = 0.426). Clinicians should be aware that younger patients who have undergone chemotherapy for breast cancer may experience changes in body image perception following surgery. Contributing factors such as younger age, use of psychiatric medications, and negative postoperative body image may be associated with a diminished quality of life related to cancer.

## 1. Introduction

Breast cancer is a condition resulting from the uncontrolled proliferation of breast cancer cells, forming tumors with the potential to invade other organs. In Brazil, it is the second most common neoplasm in women after skin cancer. In 2022, it was estimated that there would be 66,280 new cases of the disease, representing 29.7% of all oncological diagnoses in the country [1]. However, for 2023, there has been a significant increase, with estimates of 73,610 new cases resulting in an incidence rate of 41.89 cases per 100,000 women. The breast cancer mortality rate in 2020 was 11.84 deaths per 100,000 women, with the highest rates recorded in the Southeast and South regions [2].

It is essential to highlight that although breast cancer is no longer considered a death sentence due to new treatment technologies, the majority of patients still face the impact of the diagnosis and the side effects of therapies on their mental and physical health [3]. Therefore, it is crucial to consider that when a patient is diagnosed with breast cancer, it inevitably triggers a significant emotional impact. The patient may experience a series of concerns, such as fear of death, anguish, anxiety, and, in some cases, even symptoms of depression [4,5]. The diagnosis of the disease may be considered threatening and distressing for the patient and her support network since there is the possibility of mutilation of the breast and also thoughts linked to death [3,4,6]. Surgery is the primary modality for breast cancer treatment; the majority of post-surgery patients experience some degree of body image disturbance and impaired quality of life [5].

As survival rates increase, there is growing concern about the quality of life of surviving patients. Cancer treatment may have a negative impact on the body image as perceived by patients [3], and surgeries are often linked to feelings of mutilation and the perception of impairment in women’s quality of life [5,6]. Satisfaction with body image is the subjective perception of how these patients feel about their physical appearance, shape, weight, and body function [7]; contentment concerning the visual perception of one’s own body plays a crucial role in aspects related to the quality of life. It is not just a matter of vanity but a factor that can significantly influence patients’ physical and emotional well-being. In other words, after undergoing cancer treatment, these women may face a series of symptoms, such as pain, fatigue, fear of death, changes in body appearance, and impact on femininity and sexuality [8]. Additionally, side effects of chemotherapy, such as hair loss, skin irritation, change in skin color, and decreased libido, can further exacerbate these concerns. As a result, changes in appearance may be linked to reduced satisfaction with body image and quality of life [9]. Recently, a systematic literature review [3] of 51 empirical articles summarized existing knowledge about the relationship of perceived body image with the quality of life of women who have undergone surgical treatment for breast cancer. Evidence indicates that breast cancer surgery affects patients’ body image perception and quality of life worldwide. Age, education, socioeconomic status, and the type of surgery are potential factors influencing these outcomes [3].

Faced with these significant changes, research in body image has expanded to understand how the disease affects patients and develop more effective interventions to help these women and improve their well-being and quality of life [10]. An early step in developing effective interventions to foster quality of life or well-being is gathering evidence of which variables are culturally relevant to be manipulated in an intervention and predict the outcomes. Despite the topic’s relevance, studies in this area are needed in the Brazilian scenario. The present study sought to achieve two primary objectives concerning women who underwent surgical treatment for breast cancer from 2020 to 2021 at a reference university hospital in Southern Brazil. Firstly, it aimed to assess their perceptions of body image and overall quality of life related to cancer. Secondly, it explored the relative contributions of sociodemographic and clinical variables associated with oncological treatment to perceptions of body image and cancer-related quality of life. Finally, the study also investigated the specific impact of body image perceptions on quality of life while controlling for other relevant sociodemographic and clinical factors.

## 2. Methods

### 2.1. Participants

One hundred six women, breast cancer patients aged 21–93 years, were recruited from August 2022 to July 2023 via the hospital’s Mastology outpatient clinic at a large public hospital in southern Brazil (*M*_age_ = 55.38, *SD* = 12.93). The following inclusion criteria were considered for this study: patients diagnosed with breast cancer undergoing surgical treatment followed up at the hospital’s Mastology outpatient clinic between 2020 and 2021, patients over the age of 18 and with signed informed consent. Exclusion criteria were male patients, patients who have not undergone surgical treatment, women diagnosed with metastatic disease and/or undergoing palliative treatment, or who have difficulties understanding the questionnaires.

Descriptive summaries of demographic and health characteristics are presented in Table 1.

### 2.2. Variables and Instruments

*Sociodemographic data:* An ad hoc questionnaire developed for the study collected information about the participant’s age (years), education, occupation, religion, marital status, children (yes/no), addiction behavior, psychological counseling received (yes/no), and use of psychiatric medication (yes/no).

*Medical data and clinical variables related to oncological treatment:* Information about the treatment, such as disease staging, time since diagnosis, type of treatment, or surgery (conservative, mastectomy, reconstruction), was accessed from the patient’s hospital records.

*Perceived body image*: It was assessed with the Body Image and Relationships Scale (BIRS) developed by Hormes et al. [11], with cross-cultural validation for Brazil [12,13]. It evaluates attitudes about appearance, health, physical strength, sexuality, relationships, and social functioning of women undergoing breast cancer treatments [12]. The instrument consists of 32 statements, subdivided into three domains: (1) “Strength and Health,” which comprises 12 questions addressing energy to carry out physical tasks, physical strength, healthy body, and physical shape; (2) “Social Barriers’’ which includes nine questions addressing the restriction of social activities as a result of symptoms triggered by treatment (e.g., hot flashes), restriction of social activities due to changes in physical appearance, and restriction of activities social due to the physical discomfort and shame caused by breast cancer treatment; (3) “Appearance and Sexuality” which contains 11 questions on body appearance and body integrity, sexuality, and sexual desire, embarrassment, and shame with body appearance. A five-point Likert scale assesses the perception of body image in each item using an ordinal classification from 1 for “strongly disagree” to 5 for “strongly agree.” Some of the statements in the BIRS reflect a positive view of body image, while others have a negative connotation. To calculate the overall scale result, the items expressing a positive view of body image had their scores reversed. Higher scores on this scale (sum of all items) indicated worse self-evaluation, impairment, or decline in body image [13]. The internal consistency (Cronbach’s alpha and McDonald’s omega) for the complete scale obtained in this study’s sample was 0.920 and 0.923, respectively. The descriptive statistics (Min.–Max., Mean, SD) for the BIRS in the current sample are presented in the Appendix A.

*Quality of life:* The Functional Assessment of Cancer Therapy for Breast Cancer (FACT-B, version 4), developed by Coster, Poole, and Fallowfield [14] and validated for Brazil [15], was used to assess quality of life. FACT-B assesses the cancer-related quality of life through 36 questions, 27 of which refer to the general quality of life, 9 to breast cancer-specificities, and 4 to arm mobility. It has five domains: (1) “Physical Well-being,” which comprises seven items addressing physical strength and health issues in general (e.g., feeling sick); (2) “Social/Family Well-being,” which contains seven items about emotional support from their families and friends; (3) “Functional Well-being,” which has seven items about being able to work and enjoy life; (4) “Emotional Well-being,” includes six items about the mood of patients, and (5) “Additional Concerns,” with ten items about arm morbidity and appearance in patients who have undergone surgery for breast cancer treatment. It uses a format response of a 5-point Likert scale, ordinal classification, from 1 for “not at all” to 5 for “very much.” The higher the score, the better the quality of life. A psychometric study for the Brazilian population presents evidence for the validity, reproducibility, and reliability suitable for use in research [15]. The internal consistency (Cronbach’s alpha and McDonald’s omega) for the complete questionnaire obtained in this study’s sample was 0.900 and 0.901, respectively. The descriptive statistics (Min.–Max., Mean, SD) for the FACT-B in the current sample are presented in the Appendix A.

### 2.3. Procedures

The data in this cross-sectional study are part of a larger project examining the psychological aspects related to the surgical treatment of women with breast cancer. The project was approved by the Research and Ethics Committee of the hospital where the data were collected. All participants were informed about the purpose of the study, and they entered into the study only after their agreement and providing written informed consent. Eligible patients were identified through the record of surgeries performed by the Mastology service in 2020 and 2021. From January 2020 to December 2021, 487 surgical appointments occurred. However, due to the COVID-19 pandemic, only the most severe cases underwent surgery. So, 146 patients underwent surgery: 61 from March to December 2020 and 85 women from January to December 2021. Based on the nonprobabilistic sample method and the estimate of 146 surgery cases, the sample size was calculated using an online tool (Sample Size Calculator). According to the sample calculation, assuming a sampling error of 5% and a confidence level of 95%, 106 women were needed to carry out the study. The inclusion criteria for the research were women aged 18 and over, with a histological diagnosis of breast cancer, who underwent surgical treatment between January 2020 and December 2021, and who agreed to participate in the study; as exclusion criteria, patients diagnosed with metastatic disease, patients who showed difficulties in understanding and answering the questionnaires, and patients who did not agree to the use of their data for research, were not interviewed. The first contact with patients occurred via a telephone call by a trained psychologist via an institutional phone number. Three contact attempts were made on different days and times, through which the patients were invited to the study. The researcher offered two options for participation: an in-person interview or a face-to-face online interview. All participants chose a virtual format interview. The appointments were made via video conference (Google Meet); initially, the digital consent form was presented and signed (a copy was also delivered by email), and then the researcher applied the assessment protocol in an interview format. All video conferences were delivered in the institution’s private research room. The interviewer tabulated patients’ responses in the SurveyMonkey platform, where all individual data from this research were stored.

### 2.4. Data Analysis

We analyzed the data in four stages using the Statistical Package for Social Science (SPSS) software, version 25 for Windows (SPSS, Inc., Chicago, IL, USA). All variables studied (e.g., sociodemographic and clinical characteristics, questionnaire scores) were statistically described in the first stage. In the second stage, we looked for sociodemographic (e.g., age, occupation, education, marital status, having children, and having a religion) and clinical characteristics (e.g., cancer staging; time since diagnosis, type of surgery—conservative, mastectomy, reconstructive, and chemotherapy, and radiotherapy) associations with perceived body image (BIRS general score and subscales) and quality-of-life (FACT-B general score and subscales) scores. Following the recommendation in the literature [16], we used the Shapiro–Wilk test to assess data normality, given its robustness. The choice between parametric and nonparametric tests was based on the test results and the characteristics of the variables. When pertinent, the effect size of the comparisons that showed statistically significant differences was presented, following Cohen’s parameters [16]. In the third stage, correlation analyses were carried out with FACT-B and BIRS scores (domains and general scores). Then, we looked for associations between general scores and clinical and sociodemographic variables. This procedure was also used to select the variables that would be presented as independent variables in the regression models in the next stage. Pearson’s correlation coefficients (r) were used for correlations between variables with a normal distribution and Kendall’s correlation coefficient (τ) for nonparametric variables. Finally, in the last stage, two multiple linear regressions were carried out to understand which variables would be statistical predictors of body image and quality of life. The second regression analyses presented in the study were performed using the hierarchical method. The model employed a two-step approach: in the first step, the variables age and use of psychiatric medication (no = 0, yes = 1)—sociodemographic and clinical factors with theoretical relevance and a statistically significant correlation with the outcomes—were included as the initial predictors. In the second step, the remaining variables that showed statistically significant associations with the dependent variables in the preliminary analyses were added. This procedure allowed us to assess the predictive power of each group of variables while controlling for the effects of age and psychiatric medication, in line with current methodological recommendations [16].

## 3. Results

### 3.1. Associations of Sociodemographic and Clinical Characteristics with Perceived Body Image (BIRS) and Quality-of-Life (FACT-B) Scores

Most of the sociodemographic characteristics were not associated with perceived body image (BIRS) or cancer-related quality-of-life (FACT-B) scores, except for the variable age. Age was associated with the BIRS general score (*r* = −0.230; *p* = 0.020) and with the FACT-B general score (*r* = 0.314; *p* = 0.001). Appendix A provide detailed descriptive statistics and show nonsignificant associations.

On the other hand, a few medical/clinical variables were associated with perceived body image (BIRS) or quality-of-life (FACT-B) scores. For example, alcohol and tobacco use were associated with higher scores in the perception of social barriers (BIRS) (*p* = 0.001), which means experiencing more restriction of social activities due to symptoms triggered by treatment or due to changes in physical appearance or due to physical discomfort and shame caused by breast cancer treatment. The use of psychiatric medications was associated with lower physical well-being (FACT-B) (*p* = 0.049), lower emotional well-being (FACT-B) (*p* = 0.009), lower functional well-being (FACT-B) (*p* = 0.016), and higher additional concerns (FACT-B) (*p* = 0.046). Patients who had received psychological counseling showed higher scores in the domain Appearance and Sexuality of the BIRS, which means worse perceptions of body appearance and body integrity, sexuality and sexual desire, embarrassment and shame with body appearance (*p* = 0.002). Patients treated with chemotherapy showed higher scores in the BIRS general score (*p* = 0.002), which means a negative body image perception. Also, those treated with chemotherapy reported higher scores in the domain of Social Barriers of the BIRS, which means more perception that social activities are restricted or impaired (*p* = 0.000). Women treated with chemotherapy also presented higher scores in Strength and Health (BIRS), which means lower energy to perform physical assignments, physical strength, a healthy body, and physical fitness (*p* = 0.007). Appendix A, provides details concerning the descriptive statistics and statistical comparisons.

No statistically significant differences in the perceived body image (BIRS) and quality-of-life (FACT-B) scores by the time since diagnosis and the type of surgery (conservative, mastectomy, or reconstruction) were found. Appendix A, provides details concerning the descriptive statistics and statistical comparisons.

### 3.2. Associations Between Perceived Body Image and Cancer-Related Quality of Life

To verify the association between perceived body image (BIRS) and cancer-related quality-of-life (FACT-B) constructs, a correlation analysis between the general scores and dimensions of each instrument was carried out (see Table 2). Significant negative associations between instrument scores indicated that more negative perceptions of body image are associated with lower levels of perceived quality of life in the current sample (see details in Table 2).

To ensure transparency regarding the significance of the observed associations, *p*-values for all correlations are provided in Table 2. Given the number of simultaneous correlations (10 × 10 matrix), correlations with *p*-values greater than 0.001 should be interpreted with caution, as they may reflect an inflated Type I error, considering the Bonferroni correction [16].

Subsequently, two regression analyses were conducted to explore the relationships found. In the first multiple linear regression model, the dependent variable used was the perceived body image (BIRS total score), formed by the sum of the three factors of the BIRS scale [13] (Strength and Health, Social Barrier, Appearance and Sexuality). As independent variables, we considered the variables significantly correlated with the dependent variable in previous analyses (i.e., Age and Chemotherapy [no or yes]).

It is noteworthy that the assumptions of multiple regression were tested, all of which were met in this model: absence of serial autocorrelation in the residuals (Durbin–Watson: 2.01); the normality of distribution of residual values was graphically verified, as well as the presence of homoscedasticity [16], and no problems with multicollinearity were found, with no substantial correlations between predictors with all FIVs below 10 (1.004). As a result, we found a statistically significant model to predict the general perceived body image [F(2, 99) = 7.376; *p* = 0.003]. In order of relevance, chemotherapy history (standardized β = 0.277; *p* = 0.004; adjusted R^2^ = 7.5%) and age (standardized β = −0.214; *p* = 0.025; adjusted R^2^ = 3.7%) proved to be statistically significant predictors and explained 11.2% (adjusted R^2^ = 0.112) of the variance of the general perceived body image [BIRS] (See Table 3).

Next, a hierarchical multiple regression analysis was conducted to examine the factors predicting the cancer-related quality of life, as measured by the FACT-B total score (formed by summing the five facets of the FACT-B scale). Predictor variables were entered in two steps. In Step 1, age and “use of psychiatric medications” [no or yes] was entered into the model. This initial model was statistically significant, *F*(2, 99) = [7.320], *p* = 0.001, and accounted for 11.1% of the variance in general quality of life (adjusted *R*^2^ = 0.111). Age (Estimate = 0.478, *p* = 0.002) and “use of psychiatric medications” (Estimate = −8.640, *p* = 0.052) were found to be significant or marginally significant predictors of general quality of life.

In Step 2, “Strength and Health (BIRS),” “Social Barriers (BIRS),” and “Appearance and Sexuality (BIRS)” were added to the model. The inclusion of these variables resulted in a statistically significant increase in the explained variance (Δ*R*^2^ = 0.315, *F* change (5, 96) = [15.970], *p* < 0.001 [assuming significance based on Δ*R*^2^]). The full model, including all predictors, was significant, *F*(5, 96) = [15.970], *p* < 0.001, and explained a total of 42.6% of the variance in general quality of life (adjusted *R*^2^ = 0.426).

Within the final model (Step 2), age (Estimate = 0.261, *p* = 0.042) and use of psychiatric medications [no or yes] (Estimate = −8289, *p* = 0.025) remained significant predictors. Additionally, “Strength and Health (BIRS)” (Estimate = −0.435, *p* = 0.024), “Social Barriers (BIRS)” (Estimate = −0.537, *p* = 0.024), “Appearance and Sexuality (BIRS)” (Estimate = −0.750, *p* = 0.002) emerged as significant negative predictors of general quality of life. These findings indicate that while younger age and use of psychiatric medication are associated with lower quality of life, greater negative perceived body image as evaluated by challenges in physical strength and health, social interactions, and appearance and sexuality, is associated with lower general quality of life. The detailed regression coefficients, standard errors, and significance levels for both steps are presented in Table 4.

## 4. Discussion

To the best of our knowledge, this is the first study in Brazil to associate perceived body image and cancer-related quality-of-life variables in women who have undergone breast cancer surgery. We assumed in our study that the notion of body image is a complex concept that involves physiological, psychological, and social aspects that affect people’s emotions, thoughts, and the way they relate to others, influencing their quality of life and how they behave daily [3,17]. Becoming ill with breast cancer and its treatment have serious consequences that can be temporary or permanent in a woman’s life.

In our study, the average age of the participants coincides with the age group (i.e., from 40 to 60 years old) considered most susceptible to the development of breast cancer, according to the Brazilian Cancer Institute [2]. The literature indicates that younger women often grapple with more pronounced challenges stemming from treatment and surgery compared to their older counterparts (60 years or older). This affirmation coincides with the findings from our study and previous research (e.g., [3,18]) and underscores our conclusions’ importance. Specifically, our investigation revealed that younger women experienced lower body image satisfaction and quality of life, highlighting the significance of age as a crucial sociodemographic variable in the psychological assessment of women in this condition.

Although scores in the questionnaires BIRS and FACT-B were slightly favorable in those women who underwent conservative surgery compared to those who underwent mastectomy, our data did not show significant statistical differences in the scores related to perceived body image and cancer-related quality of life according to the type of surgery. A previous study [6] developed in Brazil found differences in perceived body image and quality of life comparing women who underwent radical surgery versus conservative surgery. The reasons for finding different results from our study may be that the assessment of the previous study included different instruments (i.e., Body Image after Breast Cancer Questionnaire, EORTC QLQ-C30, and QLQ BR-23) to assess the constructs. The Body Image after Breast Cancer Questionnaire [6] is a specific measure for breast cancer, which may be more accurate. Furthermore, women’s characteristics, such as age and clinical status, and the smaller sample size, may have contributed to the nonsignificant results related to the type of surgery in our research.

Furthermore, although the type of surgery is an objective variable, each treatment modality manifests distinct side effects, ranging from enduring surgical scars with immediate impact to skin alterations. Surgery perception and satisfaction may vary across individual differences; at the same time, body image and perceived quality of life are subjective variables [3]. Additionally, our study’s questionnaire to assess perceived body image (i.e., BIRS) may not encompass all the details and aspects related to body image and cancer clinical status.

Undergoing chemotherapy was another variable associated with participants’ perceived body image in the present study. Limitations in social activities may arise due to symptoms resulting from breast cancer treatment, such as changes in appearance, physical discomfort, and shame resulting from treatment [11,19,20], and side effects of medication (e.g., alopecia, loss of vaginal lubrication, chemotherapy-induced amenorrhea) can contribute to these perceptions. A systematic review [18] concluded that patients began to question the social roles they aspired to, such as professional and maternal roles, after treatment for breast cancer. These women expressed feelings of worthlessness in connection with the side effects of radiotherapy and chemotherapy, including fatigue, as well as restrictions placed on previous activities due to lymphedema. Such issues can be related to the present study’s findings on the age and chemotherapy history and their role as predictors of general perceived body image.

Almost 30% of participants were taking psychiatric medication at the time of the interview. These figures align with those reported in other studies [21,22,23]. According to clinical guidelines [22], psychological symptoms in cancer patients are often under-recognized and undertreated. Such symptoms may be dismissed as a normal response to a cancer diagnosis or seen as secondary to physical symptoms. On the other hand, many patients are hesitant to accept medication such as antidepressants [23].

Managing cancer means that a person cannot avoid coping with symptoms of anxiety and distress, sometimes related to feelings of uncertainty related to the course of the disease, daily schedules during treatment, doctor’s appointments, chemotherapy, and radiotherapy sessions [20,21]. In those cases, patients may benefit from first-line psychological or behavioral management [22]. Patients who do not have access to first-line treatment prefer pharmacotherapy, and those who previously responded well to pharmacotherapy may benefit from a pharmacologic regimen for depression or anxiety according to guidelines [22,23].

In our study, individuals using psychiatric medication demonstrated diminished physical, emotional, and functional well-being—as measured by the FACT-B—compared to those not using such medication. Although pharmacotherapy is generally expected to improve quality of life [23], clinical conditions (e.g., physical pain, hair loss, and limited arm mobility [21]), contextual stressors (e.g., frequent medical consultations), and individual differences (e.g., limited coping resources) may necessitate a more comprehensive approach. Combining empirically supported psychosocial interventions and structured physical activity with pharmacotherapy may yield better outcomes [23]. However, such interventions were not available at the hospital where the study was conducted.

Furthermore, our results indicate that more negative perceptions of body image were associated with lower quality-of-life indices, highlighting that the factors “Strength and Health,” “Social Barriers,” and “Appearance and Sexuality” partially explain the general cancer-related quality of life. The literature reports that women who underwent surgery to remove cancer had worse scores in body image dimensions and also in the quality-of-life indices [24,25,26]. The “Strength and Health” domain is related to the energy needed to perform physical tasks, a healthy body, and physical fitness. One of the reasons why this domain is associated with the overall quality of life in our study may be related to changes observed after surgery. Problems such as edema and difficulty raising their arms after breast cancer procedures are consequences for women’s physical health that hinder their ability to carry out routine activities [27,28].

It is important to recognize that a woman’s body image can be significantly affected by the loss of one or both breasts, hair loss, the presence of surgical scars, and changes in weight. These physical changes can adversely impact the overall quality of life for these women, as they may be noticeable to others [29]. This could explain why the “Social Barriers” domain is linked to the overall quality-of-life factor. The findings related to the “Appearance and Sexuality” dimension support previous studies [8,30,31], which show that a cancer diagnosis often threatens women’s sense of femininity. This threat can lead to feelings of being “less of a woman,” accompanied by shame and embarrassment over changes in their bodies. Many women worry that their partners will no longer find them attractive. For example, a qualitative study [32] of women who underwent breast cancer surgery revealed that less than one-third of the participants felt comfortable with their physical appearance. This idea corroborates with the data from our study; 29.2% (*n* = 31) of participants endorsed the BIRS item “Changes in my physical appearance that I attribute to my breast cancer surgery make me feel embarrassed.”

While this study offers valuable insights, its findings must be considered in light of several key limitations. Firstly, as with any cross-sectional design, our results preclude causal inferences. Secondly, the sample, drawn exclusively from a single public hospital in Brazil’s unified health system, comprised women predominantly from low socioeconomic backgrounds. This specificity limits the generalizability of our findings to other populations. Thirdly, our study did not account for potentially influential variables such as social support systems, body mass index, or the severity of treatment side effects, all of which could impact body image and quality of life. Finally, the Body Image and Relationships Scale (BIRS) may not fully capture the multifaceted impact of breast cancer surgery on physical appearance. Specifically, it might not adequately address issues such as scarring and aesthetic outcomes, the consequences of disease or treatment on body image (e.g., hair loss, skin alterations), or repercussions on sexual well-being (e.g., diminished attractiveness). Concerns related to breast ”amputation” and matters of identity and femininity may also not be fully captured. This limited scope may have contributed to the lack of observed differences in body image perception across surgical and treatment types.

## 5. Clinical Implications

The results of this study indicate the importance of observing changes in the perceived body image of women who have undergone breast cancer surgery, especially for younger patients and those who have undergone chemotherapy, and also those taking psychiatric medication. Thus, it is essential to consider these changes to support women during the illness and treatment process. In this context, psychological assessment plays a crucial role in guiding decisions on counseling and psychological support concerning adjustments to manage the stress triggered by treatment and the potential transformations resulting from breast cancer. In addition, using appropriate psychometric tools, accurate results of the variables assessed will help address body image and quality-of-life issues with the patient, and assist the multi-professional team to discuss these issues with the patient.

## 6. Conclusions

Our research indicates an association among factors such as age and chemotherapy treatment and women’s perceptions of overall body image following breast cancer surgery. Specifically, younger women who underwent chemotherapy reported more negative body image perceptions. The type of surgery, however, was not associated with general body image perception in our study. Furthermore, the results revealed that younger age, use of psychiatric medication, and lower body image satisfaction—including perceived reductions in physical strength, social activity limitations due to treatment, and embarrassment about physical appearance—were associated with decreased cancer-related quality of life among women post-surgery. These findings highlight the need for assessing and addressing changes in body image in women recovering from breast cancer surgery.

## Figures and Tables

**Table 1 ijerph-22-01114-t001:** Demographic and health characteristics of the sample (N = 106).

Variables		*n*	%
Age	From 18 to 39	11	10.4%
	40 to 60	51	48.1%
	61 or more	42	39.6%
	Missing data	2	1.9%
Religion	Catholic	64	60.4%
	Evangelicals	24	22.6%
	Atheists	2	1.9%
	Others/No answer	16	15.1%
Occupation	Formal work	65	61.3%
	No formal work	24	22.6%
	Retired	17	16.0%
Education	Illiterate	1	0.9%
	Complete primary education	15	14.2%
	Incomplete elementary education	28	26.4%
	Complete high school	28	26.4%
	Incomplete high school	12	11.3%
	Complete higher education	12	11.3%
	Incomplete higher education	9	8.5%
Marital status	Married	46	43.4%
	Divorced	16	15.1%
	Dating	2	1.9%
	Single	27	25.5%
	Widow	15	14.2%
Children	No	16	15.1%
	Yes	90	84.9%
Current addiction behavior	No	89	84.0%
	Yes, medicines	4	3.8%
	Yes, tobacco	13	12.3%
History of addiction behavior	No	69	65.1%
	Yes, medicines	1	0.9%
	Yes, alcohol	1	0.9%
	Yes, tobacco	35	33.0%
Use of psychiatric medications	No	75	70.8%
	Yes	31	29.2%
Had psychological counseling	No	67	63.2%
	Yes	39	36.8%
Time since diagnosis	≤ 2018	3	2.9%
	2019	12	11.3%
	2020	55	5.9%
	2021	34	32.1%
	2022	1	0.9%
	Missing data	1	0.9%
Cancer staging	Grade 1	9	8.5%
	Grade 2	41	38.7%
	Grade 3	26	24.5%
	Grade 4	7	6.6%
	Do not know	23	21.7%
Radiotherapy	No	19	17.9%
	I do not know	4	3.8%
	Yes	83	78.3%
Surgery	Conservative	69	65.1%
	Mastectomy	26	24.5%
	Reconstruction ^#^	11	10.4%
Chemotherapy	Did not	43	40.6%
	Adjuvant	32	30.2%
	Neoadjuvant	31	29.2%

^#^ The type “Reconstruction” includes the following scenarios: first, a mastectomy followed by a different surgical procedure for breast reconstruction (*n* = 5); second, a mastectomy and breast reconstruction performed during the same surgery (*n* = 2); and third, conservative surgery along with breast reconstruction in the same surgical procedure (*n* = 4).

**Table 2 ijerph-22-01114-t002:** Kendall’s correlation coefficient (and *p*-value inside the parentheses) between perceived body image scores (BIRS) and cancer-related quality-of-life scores (FACT-B) (N = 106).

Variable	1	2	3	4	5	6	7	8	9	10
1. GQL	-									
2. PWB	0.571 (0.000)	-								
3. SFWB	0.370 (0.000)	0.145 (0.037)	-							
4. EWB	0.653 (0.000)	0.387 (0.000)	0.211 (0.003)	-						
5. FWB	0.594 (0.000)	0.408 (0.000)	0.203 (0.003)	0.494 (0.000)	-					
6. AWBC	0.624 (0.000)	0.426 (0.000)	0.142 (0.039)	0.429 (0.000)	0.322 (0.000)	-				
7. GBI	−0.459 (0.000)	−0.309 (0.000)	−0.167 (0.015)	−0.356 (0.000)	−0.336 (0.000)	−0.450 (0.000)	-			
8. SH	−0.372 (0.000)	−0.269 (0.000)	−0.100 (0.145)	−0.287 (0.000)	−0.319 (0.000)	−0.337 (0.000)	0.622 (0.000)	-		
9. SB	−0.398 (0.000)	−0.268 (0.000)	−0.132 (0.056)	−0.304 (0.000)	−0.273 (0.000)	−0.409 (0.000)	0.667 (0.000)	0.359 (0.000)	-	
10. AS	−0.343 (0.000)	−0.222 (0.001)	−0.172 (0.012)	−0.318 (0.000)	−0.230 (0.001)	−0.319 (0.000)	0.601 (0.000)	0.320 (0.000)	0.390 (0.000)	-

1 = GQL: General Quality of Life [total score] (FACT-B); 2 = PWB: Physical Well-Being (FACT-B); 3 = SFWB = Social and Family Well-being (FACT-B); 4 = EWB = Emotional Well-Being (FACT-B); 5 = FWB = Functional Well-being (FACT-B); 6 = AWBC = Additional Well-being Concerns (FACT-B); 7 = GBI = General Body Image [total score] (BIRS); 8 = SH = Strength and Health (BIRS); 9 = SB = Social Barrier (BIRS); 10 = AS = Appearance and Sexuality (BIRS).

**Table 3 ijerph-22-01114-t003:** Regression model predicting the general body image (BIRS).

IV	Estimate	*SE*	95%CI		*p*
			*LL*	*UL*	
Intercept	102.637	-			0.000
Chemotherapy [no = 0, yes = 1]	12.314	0.277	4.038	20.590	0.004
Age	−0.362	−0.214	−0.677	−0.046	0.025

IV = Independent variables.

**Table 4 ijerph-22-01114-t004:** Hierarchical regression model predicting the general quality of life (FACT-B).

		Estimate	SE	95% CI		t	*p*	ΔR²
				LL	UL			
Step 1								0.111
	Intercept	72,643		54.973	90.313	8.157	0.000	
	Age	0.478	0.293	0.174	0.781	3.119	0.002	
	Use of psychiatric medication (no and yes)	−8.640	−0.185	−17.347	0.067	−1.969	0.052	
Step 2								0.315
	Intercept	135.712		113.700	157.725	12.238	0.000	
	Age	0.261	0.160	0.010	0.512	2.063	0.042	
	Use of psychiatric medication (no = 0, yes = 1)	−8.289	−0.178	−15.499	−1.079	−2.282	0.025	
	Strength and Health (BIRS)	−0.435	−0.209	−0.812	−0.058	−2.289	0.024	
	Social Barriers (BIRS)	−0.537	−0.222	−1.001	−0.073	−2.300	0.024	
	Appearance and Sexuality (BIRS)	−0.750	−0.291	−1.221	−0.280	−3.167	0.002	

## Data Availability

The raw data supporting the conclusions of this article will be made available by the authors on request.

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
