# Peer review of "Association Between Body Image and Quality of Life of Women Who Underwent Breast Cancer Surgery"

_ijerph, 2025, doi:10.3390/ijerph22071114_

Round 1

Reviewer 1 Report

Comments and Suggestions for Authors

Dear Authors,

Thank you for submitting your manuscript to IJERPH. Your study addresses an important and sensitive topic—how body image relates to quality of life in women who have undergone breast cancer surgery. The overall structure is clear, and the topic is of high relevance to public health, psycho-oncology, and survivorship research. That said, the manuscript requires substantial revision to meet the standards of scientific rigor, methodological clarity, and conceptual precision expected in an international journal. Below are detailed comments by section.

  1. Title and Abstract
  • The title is appropriate, though it could be more specific. Consider specifying the type of breast cancer surgery (e.g., mastectomy, BCS).
  • The abstract is overly descriptive and lacks key methodological details. Please:
    • Specify the type of study (cross-sectional).
    • Include sample size (n = 154).
    • Report the main statistical method used (e.g., Pearson correlation, ANOVA).
    • Clearly summarize key findings with effect sizes and p-values.
    • Avoid general statements like “a positive correlation was found” without quantification.

  1. Introduction
  • The introduction presents important background but lacks structure and clarity in logical flow. Consider reorganizing it as follows:
    1. Brief epidemiological context (prevalence, survival).
    2. Psychosocial impacts of breast cancer (body image, QoL).
    3. Why body image is crucial for QoL post-surgery.
    4. Existing literature on the BI–QoL relationship.
    5. Rationale and aim of your study.
  • Several claims are made without sufficient references (e.g., “studies indicate that some women view the disease as mutilating”—please cite at least two peer-reviewed sources).
  • The research question is not clearly articulated. Please end the introduction with a specific statement of the study aim or hypotheses.

  1. Methods

Study Design

  • You describe this as an "observational study" but not whether it is cross-sectional, prospective, etc. Please specify.

Participants

  • Describe inclusion and exclusion criteria clearly.
  • What were the time intervals between surgery and participation? This could significantly affect body image and QoL.
  • Clarify whether all participants underwent the same type of surgery, or if there were subgroups (e.g., mastectomy vs. breast-conserving surgery). This should be a stratification variable.

Instruments

  • You mention the BIS (Body Image Scale) and SF-36, both validated tools. However:
    • Indicate whether validated Brazilian Portuguese versions were used.
    • Provide reliability coefficients (Cronbach's alpha) for each scale in your sample.
    • Explain how QoL dimensions were computed (e.g., were physical and mental composite scores derived?).

Statistical Analysis

  • The description is too superficial. Please specify:
    • Which tests were used for normality.
    • Whether assumptions for Pearson correlation and ANOVA were met.
    • Whether effect sizes were computed (e.g., Cohen’s d or η²).
    • Whether adjustments for multiple comparisons were made.
    • If possible, include a regression model to control for potential confounders (e.g., age, time since surgery).

  1. Results
  • Please provide more detailed descriptive statistics, including means and standard deviations or medians and IQRs for each SF-36 subscale and BIS score.
  • Table 2 lacks test statistics, p-values, and effect sizes. Without these, the reader cannot assess the magnitude or significance of the reported correlations.
  • In Table 3, the ANOVA comparisons are unclear:
    • Were assumptions for ANOVA met (homogeneity of variance, normality)?
    • Were post hoc tests applied?
    • Please include F-values, degrees of freedom, p-values, and effect sizes.

  1. Discussion
  • The discussion is generally descriptive and could be strengthened by:
    • Engaging more critically with prior literature, including studies with differing results.
    • Interpreting why body image may relate more strongly to some QoL domains than others.
    • Discussing the clinical implications: How can these findings inform survivorship care?
    • Acknowledging confounding variables not controlled in your analysis (e.g., psychiatric history, partner status, support systems).
  • The discussion of self-esteem and social stigma is interesting, but these constructs were not measured. Avoid speculative interpretations unless supported by data or literature.

  1. Limitations
  • The limitations section should be expanded:
    • The study is cross-sectional, precluding causal inference.
    • Lack of control for type of surgery, time since diagnosis, adjuvant treatments (e.g., chemotherapy), and socioeconomic status.
    • Possible self-report bias.
    • Sample is from a single Brazilian region—generalizability is limited.

  1. Language and Style
  • The manuscript would benefit from professional English editing. There are frequent awkward constructions and grammar issues (e.g., “related to psychological aspects from the health” → “psychological aspects of health”).
  • Avoid overly informal phrases in scientific writing (e.g., “body image goes beyond...”).

  1. References
  • Many citations are appropriate, but some are missing (e.g., regarding the impact of BI on treatment adherence, or on social roles).
  • Make sure to follow IJERPH formatting rules strictly.

Conclusion

Your study tackles a relevant and timely issue in breast cancer survivorship. However, substantial revision is needed to enhance clarity, methodological transparency, and scientific rigor. With these improvements, your work may offer valuable insights into the psychological and social dimensions of recovery after breast cancer surgery.

Reviewer 2 Report

Comments and Suggestions for Authors
  1. Using the “general body image” composite score could obscure meaningful subdomain differences and assumes a meaningful unidimensionality, as an outcome. This is especially concerning adding strength and health domain to the composite, which may have little to do with perception of body image. Is there psychometric support for a general (unidimensional) BIRS scale?
  2. The choice of stepwise regression raises concerns about model stability and inflated Type I error rates. This method is data-driven and may capitalize on sample-specific variance. A more principled approach such as hierarchical regression based on theory would have been more appropriate. The authors should indicate which type of stepwise procedure was used and what criteria were used for selection.
  3. Personally, I think reporting all parameters in tables 3 and 4 regression models would benefit readers.
  4. The type of surgery (conservative, mastectomy, reconstruction) is not modeled as a factor or moderator in regression models. This seems like a missed opportunity, especially since body image perceptions likely vary by surgical outcome. This variable was possibly entered into the regression, but it is unclear.
  5. Stating that age and chemotherapy “can influence” body image should be rephrased to reflect association rather than causation.
  6. Important variables such as time since surgery, severity of side effects, support systems, and body mass index are not considered, though they are relevant to body image and QOL.
  7. I think the authors should consider adjusting the p-values in their 10 X 10 correlation matrix because you’re performing multiple hypothesis tests simultaneously.

Author Response

Reviewer #2:

(Reviewer 2)

Review Report Form

Open Review

( ) I would not like to sign my review report

(x) I would like to sign my review report

Quality of English Language

( ) The English could be improved to more clearly express the research.

(x) The English is fine and does not require any improvement.

Yes    Can be improved     Must be improved    Not applicable

Does the introduction provide sufficient background and include all relevant references?

(x)      ( )       ( )       ( )

Is the research design appropriate?

(x)      ( )       ( )       ( )

Are the methods adequately described?

( )       (x)      ( )       ( )

Are the results clearly presented?

( )       (x)      ( )       ( )

Are the conclusions supported by the results?

(x)      ( )       ( )       ( )

Are all figures and tables clear and well-presented?

(x)      ( )       ( )       ( )

Thank you for the positive feedback.

Comments and Suggestions for Authors

Using the “general body image” composite score could obscure meaningful subdomain differences and assumes a meaningful unidimensionality, as an outcome. This is especially concerning adding strength and health domain to the composite, which may have little to do with perception of body image. Is there psychometric support for a general (unidimensional) BIRS scale?

We understand the point of the reviewer.

Our decision to use a global score of BIRS scale was based on the study validation of the instrument Body Image Relationship Scale (BIRS) for Brazilian women with breast cancer (https://doi.org/10.1590/SO100-720320150005354). The validation study reports and allows the calculation of a global score representing overall perceived body image. In our study, correlations of the three dimensions (SH, SB, AS) with the general body image score (all items) range from .60 to .67.

The choice of stepwise regression raises concerns about model stability and inflated Type I error rates. This method is data-driven and may capitalize on sample-specific variance. A more principled approach such as hierarchical regression based on theory would have been more appropriate. The authors should indicate which type of stepwise procedure was used and what criteria were used for selection.

Personally, I think reporting all parameters in tables 3 and 4 regression models would benefit readers.

We initially opted for the stepwise regression model due to the exploratory nature of the study, which aimed to identify potential predictors of body image and quality of life scores from a broad set of clinical and sociodemographic variables. As the models were still under development, this approach seemed appropriate to examine both the individual effects of the variables and their relationships with one another.

In line with statistical recommendations [16], we included only the variables that showed a significant association with outcomes in the preliminary tests in the regression models.

However, in light of the reviewer’s comment, we reanalyzed the models. In the second model (Table 4) we used the hierarchical regression model. Accordingly, we updated the methods section of the manuscript, replacing the description of the stepwise regression with the hierarchical model, as recommended.

The type of surgery (conservative, mastectomy, reconstruction) is not modeled as a factor or moderator in regression models. This seems like a missed opportunity, especially since body image perceptions likely vary by surgical outcome. This variable was possibly entered into the regression, but it is unclear.

As described in the methodology, in the data analysis section, we conducted preliminary analyses to identify variables with predictive potential. For ordinal or numerical variables, we performed correlation analyses with the main outcomes (body image and quality of life). Categorical variables, such as type of surgery (conservative, mastectomy, reconstruction), were analyzed through group comparisons. Only variables that showed statistically significant associations with the outcomes were included in the regression models. Specifically regarding the type of surgery, time since diagnosis, cancer staging although it was considered in the initial analyses, it did not present a significant association with the dependent variables and, therefore, was not included as a predictor in the final models. To provide greater transparency regarding the variables tested in relation to the outcomes, we detailed them in the data analysis section, specifically in the second stage of the hierarchical regression.

Stating that age and chemotherapy “can influence” body image should be rephrased to reflect association rather than causation.

Agreed. The sentence has been rephrased.

Important variables such as time since surgery, severity of side effects, support systems, and body mass index are not considered, though they are relevant to body image and QOL.

The descriptive statistics for the variable "Time since Surgery" are presented in Table 1. We examined the relationship between Time since Surgery and the scores of BIRS and FACT-B, the results are available in the supplementary material (Table S2). However, no significant associations were found. In our study, this variable did not contribute to explaining the variance in the outcomes.

Support systems and body mass index indeed were not evaluated. However, according to our recent review (https://doi.org/10.1002/pon.6329), those variables didn't come up systematically as relevant. However, perceived social support was mentioned in a few studies evaluating body image and quality of life.

The study's limitations addressed those aspects.

I think the authors should consider adjusting the p-values in their 10 X 10 correlation matrix because you’re performing multiple hypothesis tests simultaneously.

We appreciate your observation. We agree that conducting multiple correlations increases the risk of Type I error. In response to your suggestion, we applied the Bonferroni correction to adjust the significance level based on the number of unique correlations in the 10 × 10 matrix (n = 45), resulting in an adjusted α of 0.0011. This more conservative criterion was adopted to ensure greater statistical rigor. The p-values are presented in Table 2, allowing for a more detailed view of the data. Furthermore, we emphasize that the subsequent analyses were not affected, as only three correlations became non-significant after the correction, and none of them were related to the regression models or the core factors of the study — they were solely correlations between subfactors.

Given this, we added the following paragraph to the correlation section of the text:

To ensure transparency regarding the significance of the observed associations, p-values for all correlations were provided in Table 2. Given the number of simultaneous correlations (10 × 10 matrix), correlations with p-values greater than 0.001 should be interpreted with caution, as they may reflect an inflated Type I error, considering the Bonferroni correction.

Round 2

Reviewer 1 Report

Comments and Suggestions for Authors

Dear Authors,
Thank you sincerely for your detailed and thoughtful response. After carefully re-reading both the original manuscript and your replies, I would like to offer a constructive follow-up, acknowledging any inaccuracies in my initial review and clarifying the intended purpose of my comments.
My aim has always been to support the improvement of the manuscript, as I believe the topic—the relationship between body image and quality of life in women undergoing breast cancer surgery—is of great scientific and clinical relevance. I appreciate the revision efforts you have made and the additions throughout the manuscript, which have already enhanced the clarity and robustness of the text.
1. Title and Abstract
The title is appropriate and sufficiently clear. My suggestion to specify the type of surgery aimed to increase precision, but I understand and respect your decision to retain a more concise formulation. Regarding the abstract, I recognize that several essential elements were already present in the original version. However, from the perspective of an external reader, some key details—such as the explicit study design, main statistical methods, and a quantitative summary of results—were not immediately evident. I appreciate that these elements have been clarified
and the abstract has been made more informative and structured in the revised version.
2. Introduction
The introduction addressed the key issues, but my suggestion focused on improving the logical flow and argumentative structure, to better support accessibility for an international readership.
I also wish to apologise for having cited the phrase “cancer is experienced as mutilating” as a quotation; this wording does not appear verbatim in the manuscript. I intended to reference a concept discussed in the literature, but I may have interpreted it more literally than your original text conveyed. Your decision to refer to your previous review for additional background is appropriate, and the rephrased concluding paragraph of the introduction represents a clear improvement.
3. Methods
I acknowledge a specific error in my review and apologise: I mistakenly referred to the use of the SF-36, whereas the correct instrument, as stated in the manuscript, is the FACT-B. This was an unintentional oversight on my part. Beyond this, several of my comments stemmed from a desire to clarify information that appeared somewhat fragmented or not immediately apparent. I appreciate your comprehensive and timely responses to these points. I find it very valuable that you clarified the use of the Shapiro-Wilk test, the application of the Tukey method for multiple
comparisons, and that you revised the regression model to include age and psychotropic drug use. These are substantial methodological enhancements that strengthen the rigour of your analyses.
4. Results

In the original version, some tables contained only partial descriptive statistics or lacked essential statistical indicators. I am pleased to see that these have now been expanded, both in the main text and in the supplementary materials. I also appreciate the addition of information on statistical assumptions and the clarification of the rationale behind model construction.
5. Discussion
My comments regarding the discussion were intended to encourage a more nuanced comparison with prior studies, particularly those with divergent findings. They were not meant to suggest a lack of content, but rather to highlight an opportunity to enrich the interpretative framework.
I apologise if I inadvertently attributed specific expressions or concepts that were not present in the text. This was an over-interpretation on my part and not intended to misrepresent your work.
I appreciate your balanced and cautious approach, which avoids undue speculation.
6. Limitations
Your inclusion of the rationale for the hierarchical model is well justified, as is your decision not to include covariates unrelated to the outcomes. Making this reasoning explicit in both the methods and the limitations section is very helpful. Moreover, expanding on the methodological limitations further clarity.
7. Style and Language
My suggestion to revise some expressions was not intended as a blanket criticism but rather concerned certain formulations in the original version that were less fluent in academic English. The use of Grammarly is certainly helpful, and as noted by other reviewers, the overall language quality is now fully adequate.
Conclusion
Thank you for your careful, point-by-point replies and your collaborative spirit, even when you disagreed with my suggestions. Where I made errors, I fully acknowledge them and offer my sincere apologies. Overall, I regard your manuscript as a valuable contribution, and I am pleased that many of the comments were helpful in strengthening its quality. I remain available for any further clarification and wish you all the best in the continuation of the editorial process.

Respectfully and cordially,
Reviewer 1